# Characterization of Aspartic Proteases from *Paracoccidioides brasiliensis* and Their Role in Fungal Thermo-Dimorphism

**DOI:** 10.3390/jof9030375

**Published:** 2023-03-19

**Authors:** Rafael de Souza Silva, Wilson Dias Segura, Reinaldo Souza Oliveira, Patricia Xander, Wagner Luiz Batista

**Affiliations:** 1Departamento Microbiologia, Imunologia e Parasitologia, Escola Paulista de Medicina, Universidade Federal de São Paulo, São Paulo 04023-062, SP, Brazil; 2Departamento de Ciências Farmacêuticas, Instituto de Ciências Ambientais, Químicas e Farmacêuticas, Universidade Federal de São Paulo, Diadema 09913-030, SP, Brazil

**Keywords:** aspartic protease, dimorphism, *Paracoccidioides*, paracoccidioidomycosis, PbSap, yapsin

## Abstract

Paracoccidioidomycosis (PCM) is the most prevalent systemic mycosis in Latin America and is caused by fungi from the *Paracoccidioides* genus. The infection begins after inhalation of the fungal propagules and their thermo-dimorphic shift to yeast form. Proteases play an important role in the host invasion process and immune modulation in many pathogenic microorganisms. Aspartyl proteases are virulence factors in many human fungal pathogens that play an important role in the host invasion process morphogenesis, cellular function, immunity, and nutrition. In the present study, we characterized the modulation of acid proteases from *Paracoccidioides brasiliensis*. We detected four aspartyl proteases in *P. brasiliensis* with high homology to aspartic protease from *Saccharomyces cerevisiae* Pep4. Furthermore, we demonstrated that Pepstatin A can inhibit dimorphic switching (mycelium→yeast) in *P. brasiliensis*. In addition, these genes were modulated during thermo-dimorphism (M→Y transition) in the presence or absence of carbon and nitrogen sources and during growth at pH 4 during 24 and 48 h. We also observed that *P. brasiliensis* increase the secretion of aspartic proteases when cultivated at pH 4, and these acid proteases cleave BSA, collagen, and hemoglobin. These data suggest that aspartyl proteases are modulated by environmental conditions and during fungal thermo-dimorphism. Thus, this work brings new possibilities for studying the role of aspartyl proteases in the host–pathogen relationship and *P. brasiliensis* biology.

## 1. Introduction

Yeast infections have increased dramatically in the last decade. It is estimated that 1.2 billion people in the world suffer from fungal diseases, with many suffering from chronic or invasive forms, making diagnosis and treatment difficult [1]. In addition, around 1.5 to 2 million people die each year due to fungal infections [1,2]. In Latin America, the diversity of microenvironments, climates, and habitats is responsible for creating a high and diverse population of microorganisms, including fungi. Fungi are responsible for endemic mycoses such as Coccidioidomycosis, Histoplasmosis, Sporotrichosis, and Paracoccidioidomycosis, which all have a significant impact on public health [3,4]. The increase in the incidence of cases could be related to deforestation, soil disruption, the opening up of new agricultural frontiers, and climate change due to global warming [5].

*Paracoccidioides* spp. are thermally dimorphic fungi with a broad distribution in Latin America; they are the causative agent of paracoccidioidomycosis (PCM), with the most significant number of cases reported in Brazil [6,7]. The ability of *Paracoccioides* to cause disease relies on its thermo-dimorphic feature to switch from mycelium (infective form) to yeast (pathogenic form) (M→Y) at the site of infection and adaptation to survive in the host, along with insufficient host immune response [8,9]. The mechanisms used to perform dimorphism, survive in the host, and evade the immune response have been studied, and several approaches have been carried out [8].

The development of PCM depends on interactions between fungal and host components, and proteases have been described as important factors implicated in the invasion process of *P. brasiliensis* and the fungus’s ability to cause disease [10]. Proteases are a diverse and ubiquitously distributed group of enzymes that are important for a variety of biological processes, including adhesion, cellular function, protein degradation for cell nutrition, digestion of unwanted proteins, hormone maturation, enzyme activation, immunity, morphogenesis, fertilization, and other cellular processes; in this way, they are involved in post-translational regulation, pathogenesis, and development [11,12,13,14,15].

The aspartyl proteases, otherwise known as aspartic peptidase (E.C.3.4.23.X), are enzymes from the pepsin family (A1 family) and experience activation and activity at an acidic pH. They serve many roles, but in particular they have been implicated in the virulence and pathogenicity of microorganisms such as *Candida* spp., HIV, *Plasmodium*, and *Paracoccidioides* [10,16,17,18,19,20]. Fungi in mycelium or yeast forms produce two varieties of aspartyl proteases: extracellular and intracellular. The secreted aspartyl protease (Sap) family has been extensively studied in *Candida* spp. and is considered an important virulence factor [20,21,22]. The second type of extracellular aspartyl proteases are Yapsins (Yaps), which contain a GPI moiety in the C-terminal region. Through this moiety, Yaps bind to the plasma membrane in the fungal cell, apparently implicated in the assembly and or remodeling of the cell wall and the acquisition of nutrients; in this way, some of them are virulence factors [22,23,24]. Vacuolar aspartyl proteases are the aspartic peptidases directed to the lysosomal vacuole, and pseudo vacuolar aspartyl proteases are involved in activating carboxypeptidases and other enzymes; these mature vacuolar enzymes play a pivotal role in cell survival during the nitrogen starvation and sporulation process. These enzymes can also be secreted [25,26,27,28].

In *P. brasiliensis*, a secreted aspartyl protease (PbSap) was characterized, and the protein was detected in the fungal cell wall and the culture supernatant [29]. The *PbSAP* transcript was positively expressed in the yeast form [16,30]. Quantitative proteomic analyses using the same *P. brasiliensis* isolate (Pb18) with different degrees of virulence showed that PbSap was differentially expressed in the virulent isolates, suggesting that it is a potential virulence factor [31]. Our group showed that PbSap-immunized mice experienced decreased disease progression. In addition, we observed a significant reduction in fungal infection when infected mice were treated with an aspartyl protease inhibitor, [16]. Furthermore, the gene expression of the aspartic protease PbSap is regulated during heat, oxidative, and osmotic stress [16,17], and the expression of this gene was upregulated in macrophages infected by *P. brasiliensis* yeasts [32].

Although studies on several pathogenic fungi suggest the role of proteases in pathogenesis, the aspartyl proteases are not completely understood in *P. brasiliensis* biology. Detailed analyses of aspartyl proteases in *P. brasiliensis* might enable us to determine the proteases that this fungus produces and examine the production of these enzymes in several nutritional conditions and during the dimorphic transition of yeast to mycelia, which is considered an important factor in the pathogenicity of this fungus.

## 2. Materials and Methods

### 2.1. Fungal Isolate and Growth Conditions

*P. brasiliensis* isolate Pb18 was grown in a yeast peptone dextrose-modified medium (mYPD) (0.5% yeast extract, 1% peptone, and 0.5% glucose, pH 6.7) for 4 to 5 days at 37 °C and shaking at 110–150 rpm. For mycelium growth, viable yeast cells were cultivated in mYPD or F-12 medium (Gibco^®^, Waltham, MA, USA) (MM, minimum medium) at 25 °C for 7 days at 110–150 rpm. *P. brasiliensis* isolate Pb18 was grown in RPMI 1640 for the broth microdilution assay or YNB medium (yeast nitrogen medium without amino acids) in the presence or absence of dextrose 2% and peptone 2% for RNA extraction. Viability was assessed using Trypan blue 0.4% counting on Neubauer’s chamber, using the formula: cell viability (%) = viable cells/total cells × 100. After five passages on a solid medium, the virulence of *P. brasiliensis* was recovered. Pb18 isolate was used to infect mice (B10.A) and was then reisolated, as described by Castilho et al. [17]. This study was approved by the Ethics Committee on Animal Use (CEUA-UNIFESP) of the Federal University of São Paulo, Brazil, under protocol number CEP 8888301117. Unless otherwise mentioned, all chemicals were purchased from Sigma-Aldrich (St. Louis, MO, USA).

### 2.2. Pepstatin A Inhibitor Susceptibility

The minimum inhibition concentration (MIC) was evaluated through a broth microdilution assay according to the CLSI M27-A3 standard with adaptations [33]. The antifungal activity was evaluated after the growth of Pb18 in mYPD at 37 °C and shaking at 110–150 rpm for 5 days. The cell was washed in phosphate-buffered saline (PBS) and resuspended in RPMI 1640 (1 × 10^6^ yeast cells). Yeast was incubated with different concentrations (0.01–10 µM) of Pepstatina A and Itraconazole (Sigma-Aldrich, St. Louis, MO, USA) for 7 days at 37 °C. The inhibitor was solubilized in methanol (negative control). Each yeast culture was homogenized, and 10 µL of each suspension were plated into a YPDmod agar medium. Plates were photographed after 7 days of growth at 37 °C. This assay was performed in a biological triplicate.

### 2.3. Dimorphic Transition Assay

*P. brasiliensis* yeast cells were grown in mYPD agar (pH 6.5) at 37 °C for 4 to 5 days and inoculated in a mYPD broth medium (pH 6.5). Then, yeasts were incubated at 25 °C for 7 days to reverse yeast to mycelium entirely. After the complete transition, yeasts were centrifuged at 3000× *g* and washed with PBS buffer (pH 7.2). The mycelium was grown in 6-well plates in mYPD or MM medium in the presence or absence of BSA 0.08% and 2.5 or 5 µM Pepstatin A for 5 days. Methanol (vehicle) was used as the negative control. The samples were monitored every 24 h under an optical microscope (Zeiss) at 100× magnification. This assay was performed in a biological triplicate.

### 2.4. Spot Assay

The sensitivity of Pb18 to cell wall disruptors was investigated using the spot assay. About 1 x107 yeast was incubated with Pepstatin A (2.5 µM) for 24 h at 37 °C at 150 rpm. Each yeast culture was diluted (10, 50, 100, and 500 times) in a YPDmod broth, and 7 µL of each suspension were applied to a mYPD agar medium supplemented with different cell wall-disrupting agents in the presence or absence of Pepstatin A 2.5 µM, such as Congo Red (2 µg/mL), Calcofluor White (1 µg/mL), sodium chloride (150 mM), and Menadione (10 µM). The plates were incubated for 7 days at 37 °C and then photographed. This assay was performed in a biological triplicate.

### 2.5. Paracoccidioides RNA Isolation

*P. brasiliensis* yeast cells were grown in a mYPD broth for 5 days at 37 °C with continuous shaking at 150 rpm. Cells were washed with PBS and resuspended in medium MM at a pH of 6.5 or 4 for 24 and 48 h. Fully formed yeast cells were mechanically disrupted by vortexing with glass beads for 5 min in TRIzol reagent (Invitrogen, Carlsbad, CA, USA). Pb18 growing without stimulus was used as a control for all the experiments, and fungal viability was evaluated via Trypan blue staining. RNA purification during the M→Y transition from 25 °C to 37 °C was obtained using the protocol described by Batista et al. [34]. Briefly, cell pellets of *P. brasiliensis* mycelium or mycelium undergoing the phase transition were packed in aluminum foil, frozen in liquid N_2_, and ground. The resulting powder was thawed in TRIzol. Ground cells in TRIzol were processed as suggested by the manufacturer for the purification of RNA, which was then quantified with a NanoDrop 3300 fluorospectrometer (Thermo Fisher Scientific, Wilmington, DE, USA). The quality of the extracted RNA was verified using 2% agarose gel electrophoresis, and the RNA was stored at −80 °C in H_2_O.

### 2.6. Real-Time Quantitative RT-qPCR

The abundance of *P. brasiliensis* aspartic protease transcripts for different experimental conditions was quantified using a RT-qPCR. For complementary DNA (cDNA) synthesis, 500 ng of RNA was initially submitted to the DNase I enzyme (Thermo Fisher Scientific, Waltham, MA, USA) and then to the ProtoScript First Strand cDNA Synthesis kit (New England BioLabs, Ipswich, MA, USA), according to the manufacturer’s instructions. To assess gene expression via a real-time quantitative PCR, the reaction was performed with SYBR^®^ Green Master Mix (Applied Biosystems, Foster City, CA, USA) according to the manufacturer’s instructions. The endogenous expression genes for α-tubulin (*α-TUB*) and ribosomal protein 60S L34 (*L34R*) were used as housekeeping genes. The samples were prepared in triplicate in a 96-well plate (0.2 mL MicroAmp™ Optical 96-Well Reaction Plate—Applied Biosystems) compatible with the equipment used, and the plate was sealed with an optical adhesive (MicroAmp™ Optical Adhesive Film—Thermo Fisher Scientific, Waltham, MA, USA). The equipment used was the ABI StepOne Plus Real-Time PCR System (Applied Biosystems) with the following conditions: 10 min at 95 °C, followed by 40 cycles of 15 s at 95 °C and 1 min at 60 °C. The dissociation curve included an additional cycle of 15 s at 95 °C, 20 s at 60 °C, and 15 s at 95 °C. The curves of oligonucleotide efficiency were evaluated from a cDNA obtained previously and serially diluted (100, 10, 1, and 0.1 ng/µL). The Ct values of each dilution point were determined and used to make the standard curve and, finally, to calculate the primer efficiency (E = 10( − 1⁄slope) − 1 × 100). The relative expression was determined based on the 2^−ΔΔCt^ method [35]. The sequences used for each gene are listed in Table 1.

### 2.7. Bioinformatic Analysis

Deduced amino acidic sequences from genes encoding for aspartyl proteases were obtained from the *Saccharomyces* Genome Database (SGD) (ID protein numbers; *Saccharomyces cerevisiae* PEP4 (YPL154C)). Protein domains were found by inputting sequences into the Fungi Data Base (FungiDB). Phobius predictor, an algorithm based on a hidden Markov model, was used to search for signal peptides and conserved region domains from aspartyl proteases. Additional searches were performed using the Basic Local Alignment Search Tool. The structural analyses of the gene and the translated protein were carried out on the following software programs: the peptide signal was identified through SignalIP-6.0 [36], molecular weight, isoelectric point, and interactome were calculated with Compute pI/Mw and String consecutively with Expansy tools, N-glycosylation sites were identified with NetNGlyc 1.0 [37], GPI motifs were identified through KoHGPI [38], and PDB similarity was compared in the Protein Data Bank in Europe (PDBe) [39].

### 2.8. Endoprotease Assay

*P. brasiliensis*-secreted proteins were extracted according to a protocol established by Camargo et al. [40] with modifications. Briefly, Pb18 was cultured on pH 6.5 or 4 mYPD agar medium for 5 to 7 days at 37 °C. Then, the fungal mass was recovered and weighed, and 1 mL of reaction buffer (0.1 M citric acid/sodium phosphate, pH 3.3) was added to 300 mg of fungus. The suspension was mixed for 10 s on a Vortex-mixer, with a 10-s rest on ice and 10 s on the vortex. Samples were centrifuged at 12,000× *g* for 15 min at 4 °C. The supernatant was recovered and used in the acid protease activity assays. Cellular viability was determined using Trypan blue. Proteolytic activity was measured using a modification of the method described by O’Donoghue et al. [41]. Briefly, 20 µL of the sample were added to 80 µL of 7.5 µM bovine serum albumin (BSA) or 10% gelatin in 0.1 M citric acid/sodium phosphate, pH 3.3, and incubated at 50 °C. Then, 20 µL were removed after 5, 15, 30, or 45 min and added to 180 µL of a Coomassie PlusTM protein assay reagent (Pierce) in a microtiter plate. Absorbance was read at 590 nm, and activity (IU) was calculated relative to BSA degradation using a standard curve. One IU is defined as the degradation of 1 µmol of BSA min^−1^.mL^−1^.

### 2.9. Statistical Analysis

The data contained in this work were validated with the reproducibility of at least three independent experiments. For comparison analysis, a Student’s *t*-test and significance analysis were performed, as were the one-way variance (ANOVA), followed by Dunnett’s test. Differences were considered significant when *p* ≤ 0.05.

## 3. Results

### 3.1. Identification and Characterization of Aspartic Proteases of P. brasiliensis

Four typical sequences of aspartic protease (PADG_12056, PADG_08282, PADG_00634, PADG_03432) were identified in the *P. brasiliensis* genome. We compared different aspartic proteases from multiple fungi (Appendix A) and observed that two predicted proteins (PADG_12056 and PADG_08282) were putative extracellular proteases related to both GPI-anchored with high homology to Yapsins; we named them PbYap1 and PbYap2, respectively (Table 2). In addition, the predicted protein from the PADG_00634 (PbSap, previously characterized by Tacco et al., [29] and PADG_03432 (now named PbSap2) showed very high similarity with yeast and filamentous fungi vacuolar aspartic endoproteases (Table 2). The sequence PADG_12056 was noted as “C Subunit Replication Fact”, but we detected an error in this sequence upon comparison to the same sequence of other *Paracoccidioides* isolates (such as Pb03, PABG_06949, and PABG_06947). We showed that PADG_12056 are two joined predict genes (data not shown). Then, all analyzes of the PADG_12056 amino acid sequence and primer design were performed between 1080 and 2538 bp (which corresponds the PbYap1 sequence). The sizes of the aspartic protease’s genes ranged from 1458 bp to 1601 bp (Table 2). The comparative analysis of aspartic proteases, such as Pep4 of *S. cerevisiae*, revealed a high percentage of shared identity, between 29 and 59% (Table 2), demonstrating high conservation among these proteins.

The *PbSAP*, *PbSAP2*, *PbYAP1*, and *PbYAP2* genes encode for putative proteins ranging from 400 to 495 amino acids (Table 2), with molecular weights ranging from 43.8 to 57 kDa (Table 3) and acid isoelectric points (Table 3). We observed that the conserved aspartic residues in the DTG and DXG motifs, crucial for aspartic protease activity [42,43], are present in all four typical aspartic proteases. Two other characteristic aspartic protease motifs, a group of cysteine residues involved in disulfide bond formation and a 19-21-amino acid putative N-terminal signal peptide, were detected (Table 3 and Figure 1A). PbSap and PbSap2 showed two characteristic N-glycosylation sites, while PbYap1 and PbYap2 showed five and three, respectively (Table 3 and Figure 1A). The PBD database showed that PbSap, PbSap2, PbYap1, and PbYap2 had high similarities with other aspartic proteases sequences already studied, mainly PbSap with Saccharopepsin (Table 3).

### 3.2. Interactome of Aspartic Protease Proteins of P. brasiliensis

Using the STRING software to assess protein–protein interactions (PPIs), we identified 12 prominent proteins involved with PbSap, PbSap2, and PbYap2 (Figure 1B, left panel). The protein ID PbYap1 (PADG_12056) was not used in these data because STRING does not accept modified FASTA sequences that are not in one’s database. Thus, we used the FASTA sequence of PAAG_03619, corresponding to PbYap1 of the Pb01 strain. In interactions with PbSap, we observed an important autophagy-related protein (ATG8 family ID PADG_08109) (Figure 1B and Table 4). Ribeiro and colleagues [44] showed that autophagy is triggered in *P. brasiliensis* during the thermal-induced mycelium-to-yeast transition [44]. These data suggest that aspartic proteases play a role in switching mycelium to yeast in *P. brasiliensis*. The analysis of metabolic pathways carried out in FungiDB showed that proteins identified as PADG_05225, PADG_08047, PADG_11833, and PADG_03562 proteins might be involved with the pathways of the pyrimidine metabolism and UMP (uridine 5’-monophosphate) biosynthesis, which are the precursors of the pyrimidine nucleotide (Table 4). This nucleotide can subsequently be converted to all other pyrimidine nucleotides. In addition, PADG_03562 and PADG_05608 proteins may be involved in purine metabolism (Table 4). These data suggest the involvement of aspartic proteases in important cellular processes, such as nucleic acid metabolisms. Notably, PADG_08047 and PADG_11833 proteins may be involved in drug metabolism, suggesting an important role of aspartic proteases in antifungal resistance.

### 3.3. Inhibition of Aspartic Protease Proteins Causes a Delay in the Mycelium-to-Yeast Transition

Acid proteases have been related to the morphological modulation of fungi, such as *Ustilago maydis*, *Talaromyces marneffei*, and *Candida albicans* [45,46,47]. Then, we examined the effects of aspartic protease inhibition in the mycelium-to-yeast (M→Y) dimorphic transition in *P. brasiliensis*. Initially, *P. brasiliensis* yeast cells were treated with different concentrations (0.01–10 µM) of the acid protease inhibitor (Pepstatin A), and its toxicity was evaluated. There was no impact on the viability of the fungus in the different concentrations tested (Appendix A). In addition, the protein extract of the fungus cultivated under control or pH 4 conditions was used to evaluate the activity of aspartic proteases in the presence and absence of pepstatin A (1; 2.5 and 5 μM). The aspartic protease activity was higher at a pH of 4 compared to the control condition. Furthermore, in both conditions, there was a dose-dependent reduction in proteolytic activity in the presence of pepstatin A. However, no significant reduction in proteolytic activity was observed when cultured at a pH of 6.5 or 4 in the presence of the PMSF inhibitor (Appendix A). Then, we chose the concentration of 2.5 µM of Pepstatin A for the inhibition assays. This concentration agrees with other works that use Pepstatin A to inhibit aspartic proteases [48,49].

*P. brasiliensis* yeast cells were reverted to mycelium by incubating cells at 25 °C; the complete transition was followed with microscopy. *P. brasiliensis* mycelium cells were incubated at 37 °C in the presence or absence of 2.5 µM Pepstatin A, BSA 0.008% (organic nitrogen source), or vehicle and examined via microscopy for up to 96 h. The medium was supplemented with BSA to verify whether this organic nitrogen source could modulate fungal dimorphism. At standard conditions, *P. brasiliensis* mycelium cells started reverting to yeast form after 24 h of incubation at 37 °C, with a partial transition after 96–120 h (Figure 2). The mycelium cells treated with the vehicle showed a similar transition profile (Figure 2) as the control. Interestingly, when mycelium cells were treated with 2.5 µM Pepstatin A and incubated at 37 °C, there was a delay in the transition to yeast form (Figure 2). However, the presence of BSA 0.008% (Figure 2) or growth in low pH (data not shown) during the M→Y transition did not impact the speed of change. These data demonstrate that Pepstatin A regulates dimorphic switching (M→Y) in *P. brasiliensis*.

### 3.4. Aspartic Protease Genes Are Regulated during the Dimorphic Transition, at a Low pH, and in the Presence of Different Sources of Nutrients

We evaluated aspartic protease expression during dimorphism in *P. brasiliensis*, under low pH conditions and in the presence of different sources of nutrients. First, we measured *PbSAP*, *PbSAP2*, *PbYAP1*, and *PbYAP2* transcription levels at the M→Y transition. We observed a very similar gene expression profile between genes. During the M→Y dimorphic transition, we observed a 2–3.5-fold increase in gene expression. The exception was the 72 h points of *PbSAP* and *PbYAP1* (Figure 3), which decreased expression and then increased again at 96 h.

Aspartic proteases are more active at an acidic pH [50], so we evaluated the gene expression in yeast cells cultured at a pH of 6.5 (control condition) or a pH of 4 for 24 and 48 h. All aspartic protease genes significantly increased when cultivated at a pH of 4 for 48 h (Figure 4A). We did not observe early gene modulation under the conditions tested, except for the *PbSAP2* gene (Figure 4A). These data indicate that these genes are responsive to a pH of 4 in *P. brasiliensis*.

Extracellular proteinases are secreted primarily for nutrient acquisition and essential for acquiring carbon and nitrogen sources [50], as seen in the saprophytic fungi *Aspergillus niger* and *Neurospora crassa* [51]. In addition, fungal-secreted proteinases are directly involved in tissue adhesion and invasion [52,53] and in modulating the host immune system to prevent its antimicrobial action [54]. Then, we evaluated the expression of aspartic protease genes of yeast cells cultured in a YNB medium (pH 6.5) in the presence or absence of 2% dextrose and 2% peptone for 48 h at 37 °C. The gene expression of aspartic proteases was upregulated regardless of the analyzed condition. *PbSAP* and *PbYAP2* gene levels were significantly modulated in the absence of carbon and nitrogen sources by approximately 2.5 times compared to the control (Figure 4B). In these genes, similar expression levels were observed when the fungus was grown with only one nitrogen source (Figure 4B). Only one carbon source was also able to induce gene expression, but at lower levels (up to 1.5 times) than the other conditions (Figure 4B). *PbYAP1* gene expression was significantly increased in both conditions—in the starvation condition (3.3-fold) and in the presence of peptone (7-fold) (Figure 4B). On the other hand, the carbon source (dextrose) did not induce its gene expression (Figure 4B). *PbSAP2* gene expression levels increased in all conditions tested, showing an eight-fold increase in starvation conditions compared to the control (Figure 4B). These results indicate that aspartic protease genes are upregulated in starvation and independent of the presence of a carbon or nitrogen source, except for the *PbYAP1* gene, which did not increase its expression in the presence of a carbon source.

### 3.5. Role of Aspartic Proteases in P. brasiliensis Cell Wall Maintenance and Response to Oxidative Stress

The implication of fungal Yapsins in cell wall assembly and/or remodeling studies performed on *S. cerevisiae* and *C. albicans* confirmed their importance for cell wall integrity [24,55,56]. In addition, the transition from mycelium to yeast triggers the cell wall morphogenesis machinery, which involves synthesizing several cell wall sugars and proteins critical to survival during infection and the evasion of the immune system [57]. Previously, our group demonstrated a modulation in *PbSAP* expression after oxidative and osmotic stress [16]. Therefore, the spot assay was performed to evaluate the response to different types of stress. *P. brasiliensis* yeast cells were cultured in a YPDm medium (pH 6.5) supplemented with different stressors in the presence or absence of Pepstatin A. We observed that the control yeast cells grown in the presence of Pepstatin A showed a delay in growth (Figure 5). In addition, yeast cells exposed to CR, Menadione, SDS, and NaCl showed reduced growth compared to the control in the presence of Pepstatin A (Figure 5). These data indicate that Pepstatin A plays a role in regulating and stabilizing the *P. brasiliensis* cell wall.

### 3.6. P. brasiliensis Secretes Aspartyl Proteases under Acidic Culture Conditions

To assess whether these proteases are being secreted and are active, an endoprotease assay was performed using the proteins secreted by fungus cultivated at a pH of 6.5 and a pH of 4. Initially, we performed an enzyme kinetics assay to detect the optimal timing of the activity of secreted aspartic proteases. This assay observed that the best time for enzymatic activity was 35 min (data not shown). Additionally, in this assay, the digestion of BSA by aspartic proteases in the secretome of *P. brasiliensis* correlated with a reduction in absorbance after Coomassie Blue staining, according to a protocol established by O’Donoghue et al. (2008) [41]. We observed that acidic endopeptidase activity was detected only in the secretome of *P. brasiliensis* cultivated under acid conditions (pH 4). This activity was inhibited in a dose–response manner in samples treated with Pepstatin A (5, 10, and 15 μM). On the other hand, no inhibition was observed in samples treated with PMSF (serine protease inhibitor) (Figure 6A). In samples of the secretome of *P. brasiliensis* cultivated under normal conditions (pH 6.5), the levels of enzymatic activity were deficient (Figure 6A).

The search for nutrients and establishment in the tissue are essential requirements for manifesting infection by pathogenic microorganisms [58]. When in the environment, fungi are trained to compete for nutrients and can adapt quickly to environmental changes [59]. In addition, predators such as amoeba and nematodes often attack fungi in their natural environment. The strategy for escaping these predators is presented as a stimulus for increasing virulence factors for human infections [60]. This way, secreted peptidases utilize structural proteins such as collagen for adhesion and nutrient acquisition [41]. In addition, they can cleave hemoglobins for nutrient acquisition and invasion [61]. Secreted aspartic proteases from *P. brasiliensis* exhibited activity with another substrate. The activity using BSA as a substrate was defined as 100%, followed by the activity on collagen and hemoglobin (Figure 6B). We observed aspartic protease activity in collagen (98%) and hemoglobin (71%) only in samples of secretome cultivated under a low pH (pH 4). Both substrates’ digestion was partially inhibited in the presence of Pepstatin A (Figure 6B). These data demonstrate that aspartic proteases are secreted and active in *P. brasiliensis* grown under acid conditions.

## 4. Discussion

The pathogenicity of fungi in humans depends on a broad range of virulence factors [62,63]. To infect humans, fungi must overcome different barriers [64]. To achieve host colonization and invasion, hydrolytic enzymes such as phospholipases, lipases, and proteases, which are secreted by the fungal cells, play crucial roles [65]. Among these enzymes, aspartic protease, which effectively degrades numerous proteinaceous targets within the host organism, constitutes an important group of fungal virulence factors [20,51,66,67,68,69]. Previously, we demonstrated that PbSap, a secreted aspartyl protease, is an important virulence regulator in *P. brasiliensis* [16,31]. In addition, fungal-secreted aspartyl proteases are reported to mediate virulence directly [21,70].

Aspartic proteases are present in diverse microorganisms and play a crucial role in nutrition acquisition, pathogenesis, and regulating cytoplasmatic pH (pHc) [68,71]. This enzyme group belongs to the pepsin family (A1 family), which usually shows better activity under acidic conditions, both extracellular and in acidic intracellular compartments such as vacuoles/lysosomes [43,72]. Aspartic proteases are monomeric enzymes with two domains, each containing essentially an aspartate residue for its enzymatic activity. Filamentous fungi and yeast produce two varieties of aspartic proteases: (i) the family of secreted aspartyl proteases (Saps), which play an important role in the virulence of Candida spp. and whose activity/function has been studied extensively [21,70], and (ii) the Yapsins (YAPs), which have a GPI fraction in their C-terminal region. Through this GPI region, YAPs can bind to the plasma membrane and the cell wall of the fungus, showing involvement with the construction or formation of cell wall remodeling [23,24,70]. In addition, both Saps and Yaps can act as virulence factors [28].

To assess the role of aspartic proteases in *P. brasiliensis*, bioinformatics analysis was performed using the Pep4 protease from *S. cerevisiae* as a model. In this analysis, four acid proteases were identified in the genome of *P. brasiliensis*, namely: vacuolar protease A, called PbSap (PADG_00634) [16,29], PbYap1 (PADG_12056), PbSap2 (PADG_03432), and PbYap2 (PADG_08282), which were named in this work. Analysis of multiple alignments demonstrated that all proteases had conserved aspartate-containing sites and disulfide bonds in the C-terminal region. In addition, the GPI anchor sequence was identified using the KoHGPI tool [38] in the aspartic proteases with IDs *PbYAP1* and *PbYAP2*. Finally, an evaluation of the interactome of PbSap1, PbSap2, PbYap1, and PbYap2 was performed and suggested that the function of these proteases is involved in the regulation and acquisition of carbon and nitrogen, autophagy, nucleic acid metabolisms, and the activation of other proteases. Particularly, the interaction of aspartic proteases Pep4 and Atg8 plays an important role in autophagy, since the absence of Pep4 (Vacuolar protease A) completely blocks autophagy [73].

Autophagy is a highly conserved catabolic pathway for degrading entire lysosome organelles and macromolecules. It is triggered by any form of cellular stress or cellular damage [74]. Recently, Ribeiro and coworkers (2018) [44] suggested that autophagy may be involved in the adaptation processes that *P. brasiliensis* requires to overcome thermal-induced dimorphism. In filamentous fungi, many studies have demonstrated the importance of autophagy for differentiation, adaptation, development, and reproduction [75,76,77,78]. In addition, there is a relationship between autophagy-related stress responses and aspartic protease activity [79,80]. In the present work, we evidenced the role of aspartic protease inhibitors during the transition from mycelium to yeast in *P. brasiliensis*. We observed that the transition from mycelium to yeast was inhibited when yeast cells were treated with Pepstatin A. Additionally, we observed that all aspartic protease gene expression was upregulated during the transition from M→Y, suggesting the existence of an important role in the fungus transition from its infective to pathogenic forms. The participation of aspartic proteases in dimorphism modulation has also been observed in other fungi [47,81,82,83]. *Ustilago maydis* cannot make the transition from the yeast to mycelial phases in the presence of Pepstatin A or when the *PEP4* gene is suppressed. This process occurs inversely in *Paracoccidioides* spp. because the pathogenic form of *U. maydis* is the mycelium, and to establish the infection, one must change from yeast to mycelium [47]. Furthermore, its transition is accelerated when the fungus is grown in a low-pH medium and supplemented with an organic nitrogen source. On the other hand, this phenomenon was not observed in *P. brasiliensis*.

We also observed that yeast cells of *P. brasiliensis* produce extracellular aspartic peptidases when grown under low-pH conditions (pH 4) and can digest several human proteinaceous substrates (e.g., albumin, collagen, and hemoglobin), being sensitive to pepstatin A. Conversely, low activity was observed in yeasts grown at a pH of 6.5 (normal growth condition). In addition, all aspartic protease gene expression increased when *P. brasiliensis* yeast cells were cultured at an acidic pH or in starvation conditions. In *Candida* spp., aspartic proteases contribute to the adhesion and invasion of host tissues by degrading cell surface structures. *C. albicans* mutants lacking *SAP4* to *SAP6* were significantly more susceptible to phagocytosis than wild-type cells. As a result of ingestion, *C. albicans* may germinate and secrete Sap4 to Sap6, which are optimally active at the same pH as that found in the phagolysosome [51]. Previous studies by our group showed that mice infected with *P. brasiliensis* treated with aspartic protease inhibitor reduced lung damage and increased macrophages’ fungicidal and phagocytic capacity [16]. However, further studies are needed to assess how the current *P. brasiliensis* aspartic proteases regulate the evasion of the immune system.

The regulation of the gene expression of aspartic proteases was evaluated in the absence of carbon (dextrose 2%) and nitrogen. An increase in gene expression of all aspartic proteases was observed in the absence of both of these factors. However, it is possible to observe a specific regulation for *PbSAP* and *PbSAP2* in the absence of both nutrients compared to *PbYAP1* and *PbYAP2*, which preferentially demonstrated an increase in their gene expression in the presence of the nitrogen source. Recently, *Talaromyces marneffei* aspartyl protease gene deletion has been shown to affect intracellular growth and reduce pathogenicity in fungus [46]. However, the individual role of each aspartic protease can be defined only using genetic manipulation techniques for the deletion (knockout) of each gene and its virulence assessment.

The stress response of *P. brasiliensis* decreased when the fungus was treated with an aspartic protease inhibitor, mainly in the presence of cell wall disruptors, suggesting that aspartic protease would participate in the cell wall integrity response. Aspartic protease seems to participate in a broad stress response mechanism and affects regulatory processes such as gene expression and programmed cell death [80]. Furthermore, Yapsins play a role in cell wall homeostasis by removing and releasing GPI-anchored cell wall proteins [84]. In addition, Yapsins have been implicated in the proper functioning of the vacuole and pH homeostasis [85].

## 5. Conclusions

In this work, we performed in silico analysis that identified four aspartic proteases in *P. brasiliensis* with conserved characteristics. Pepstatin A (an aspartic protease inhibitor) delayed fungal thermo-dimorphism and induced fungal sensitivity to different stressors, mainly cell wall stressors. Furthermore, gene expression of aspartic proteases from *P. brasiliensis* was upregulated in acid pH, starvation conditions, and during fungal thermo-dimorphism. Secreted aspartyl proteases from *P. brasiliensis* could degrade different types of proteins, such as BSA, collagen, and hemoglobin. Collectively, our data bring new possibilities for studying the role of aspartyl proteases in the host–pathogen relationship and the biology of *P. brasiliensis*.

## Figures and Tables

**Figure 1 jof-09-00375-f001:**
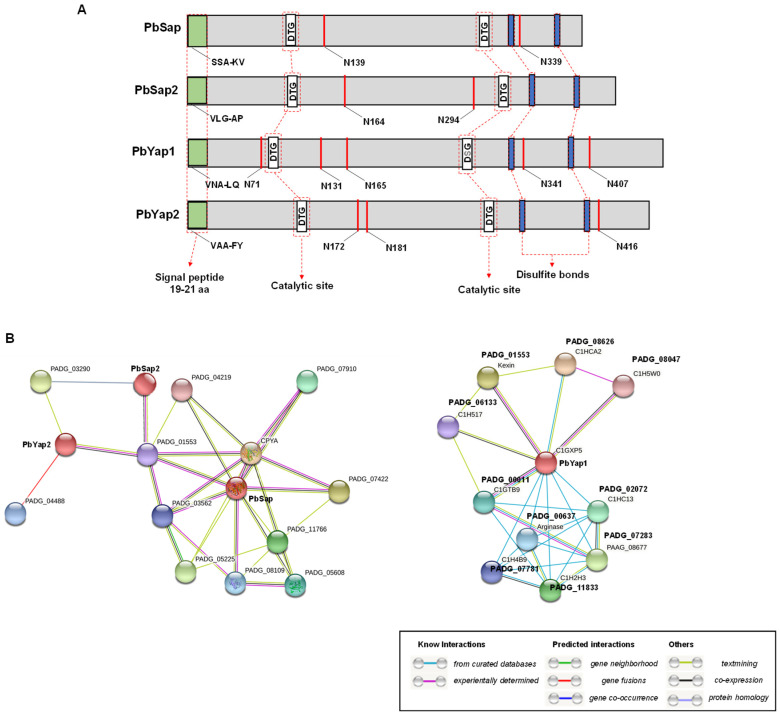
(**A**) Schematic presentation of aspartic protease of *P. brasiliensis.* Four encoded genes for aspartic protease were identified in *P. brasiliensis*: PbSap (PADG_00634), PbSap2 (PADG_03432), PbYap1 (PADG_12056), and PbYap2 (PADG_08282), using the Pep4 sequence of *Sacharomyces cerevisiae* (YPL154C) as the model. Catalytic sites (DTGSS and D(S/T)GTT) were confirmed using the Pfam database (white box). Signal peptides were identified using the tool SignalIP-5.0 (green box), and the N-glycosilatyon sites were identified through NetNGlyc 1.0 tool (red) and the cysteines (Cys) responsible for disulfide bonds (blue). (**B**) A typical association network in STRING. The aspartic proteases PbSap (PADG_00364), PbSap2 (PADG_03432), PbYap1 (PADG_12056), and PbYap2 (PADG_08282) from *P. brasiliensis* have been selected as inputs. The confidence cutoff for showing interaction links has been set to ‘highest’ (0.900).

**Figure 2 jof-09-00375-f002:**
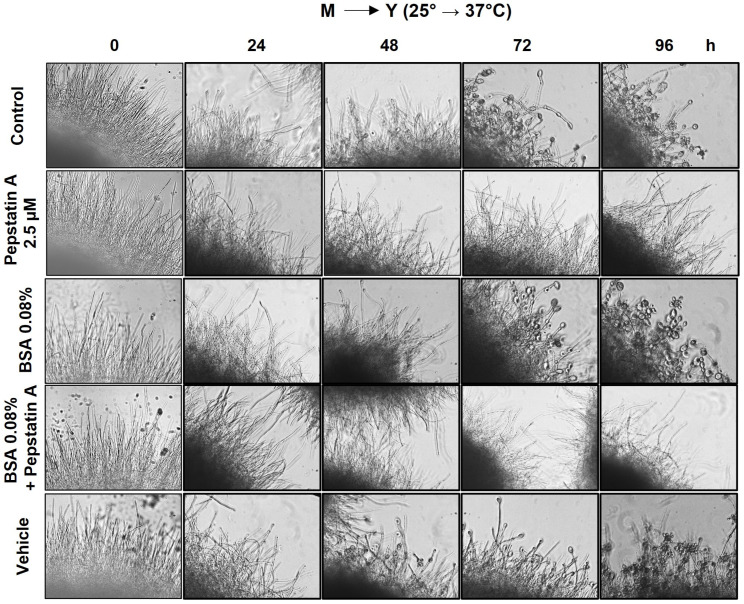
Aspartic proteases are involved in the transition from mycelium to yeast in *Paracoccidioides brasiliensis*. Mycelial cells growing in the early exponential phase were induced to undergo morphological transformation by changing the temperature of incubation. *Paracoccidioides brasiliensis* mycelium cells were treated in the presence or absence of DMSO (vehicle), Pepstatina A (2.5 μM), and BSA (0.008% *w*/*v*) and incubated at 37 °C for 4 days. After 48 h incubation, the mycelium cells started to differentiate into yeast in untreated (control) and DMSO-treated cells, but not in cells treated with 2.5 μM Pepstatin A. The results shown are representative of three independent experiments.

**Figure 3 jof-09-00375-f003:**
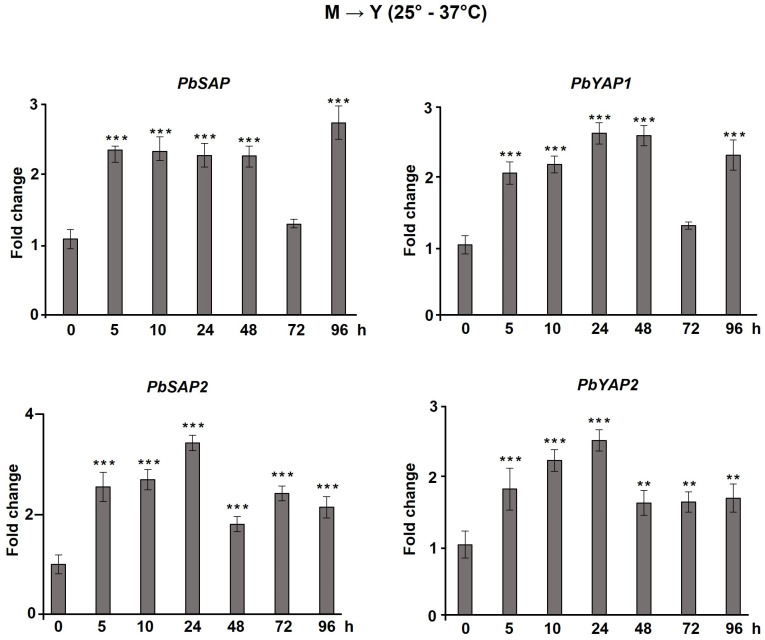
Expression of aspartic protease genes in *P. brasiliensis* during transition of mycelium to yeast (M→Y). *PbSAP* (PADG_00634), *PbSAP2* (PADG_03432), *PbYAP1* (PADG_12056), and *PbYAP2* (PADG_08282) transcript levels were measured during the mycelium-to-yeast (M→Y) transition. The change in transcriptional levels was calculated via the 2^−ΔΔCt^ method, with two housekeeping genes (*α-TUB* and *18S*). All of the data shown in this figure were analyzed using ANOVA (analysis of variance) with Dunnet’s post hoc test. Error bars correspond to the standard deviation of measurements performed in triplicate, and asterisks indicate statistically significant differences in expression (** *p* ≤ 0.01 and *** *p* ≤ 0.001). The results shown are representative of three independent experiments.

**Figure 4 jof-09-00375-f004:**
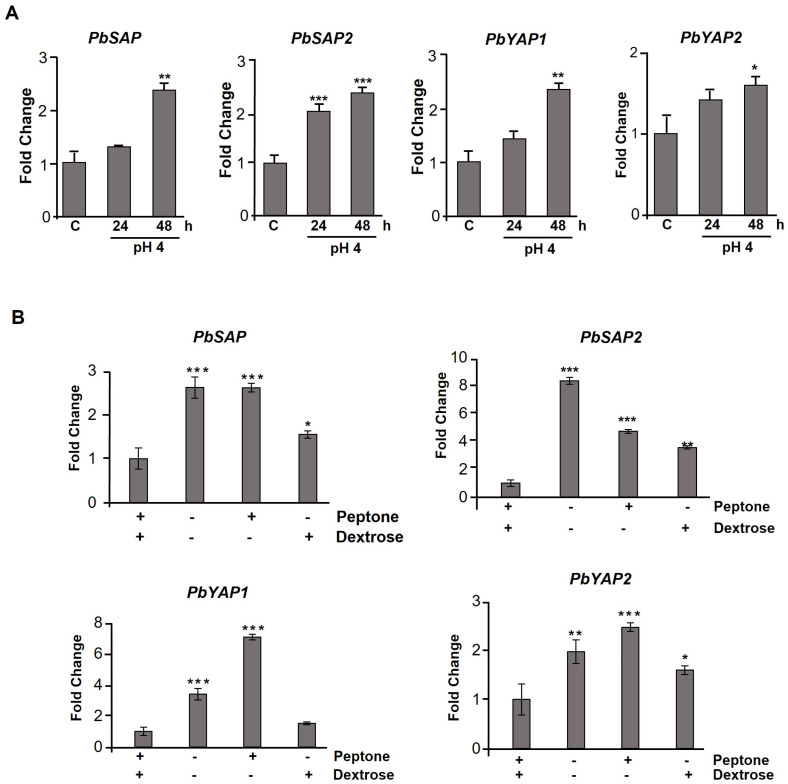
Expression of aspartic protease related genes in *P. brasiliensis*. (**A**) *PbSAP* (PADG_00634), *PbSAP2* (PADG_03432), *PbYAP1* (PADG_12056), and *PbYAP2* (PADG_08282) transcript levels were measured in yeast cells cultivated at a pH of 6.5 or 4 for 24 and 48 h. (**B**) *P. brasiliensis* yeast cells were cultivated in a YNB medium in the presence and absence of peptone (2%) and dextrose (2%), and the expression of aspartic protease genes was analyzed. The change in transcriptional levels was calculated via the 2^−ΔΔCt^ method, with two housekeeping genes (*α-TUB* and *18S*). All of the data shown in this figure were analyzed using ANOVA (analysis of variance) with Dunnet’s post hoc test. Error bars correspond to the standard deviation of measurements performed in triplicate, and asterisks indicate statistically significant differences in expression (* *p* ≤ 0.05, ** *p* ≤ 0.01 and *** *p* ≤ 0.001). The results shown are representative of three independent experiments.

**Figure 5 jof-09-00375-f005:**
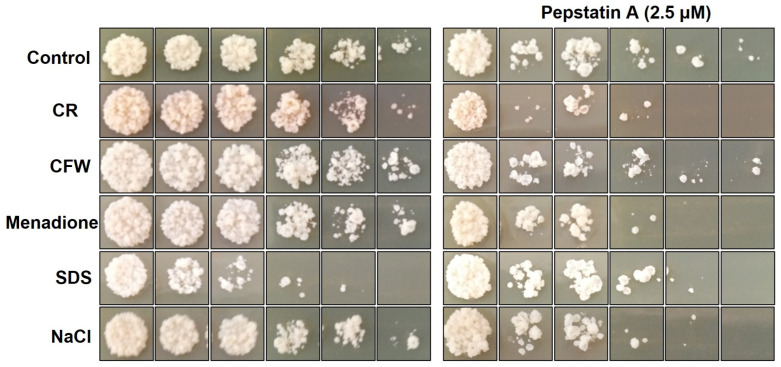
Aspartic proteases modulated the growth of *Paracoccidioides brasiliensis* cultivated in the presence of different stressors. *P. brasiliensis* yeast cells (1 × 10^6^ cells) were diluted, plated in a solid mYPD medium containing different agents that disturb the cell wall, oxidative stress, and osmotic stress, such as Calcofluor White (CFW—1 μg/mL), Congo Red (CR—2 μg/mL), Sodium dodecyl sulphate (SDS—0.001%), Menadione (10 μM), and NaCl (150 mM) in the presence or absence of Pepstatina A (2.5 µM). Finally, cells were incubated for seven days at 37 °C. The results shown are representative of three independent experiments.

**Figure 6 jof-09-00375-f006:**
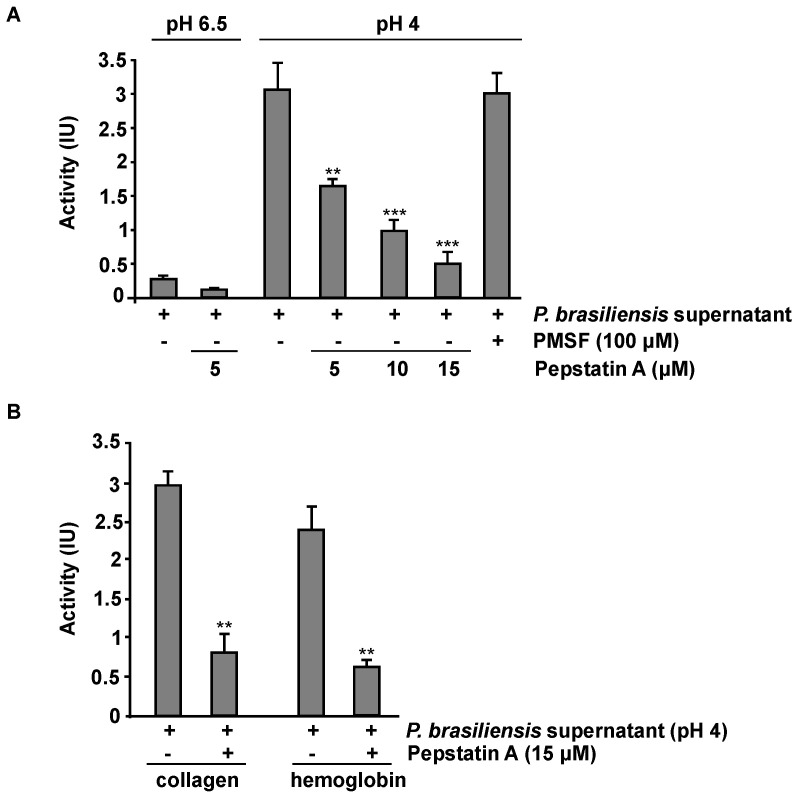
Proteolytic activity profile of secreted acid proteases from *P. brasiliensis*. (**A**)Yeast cells were grown in YPDm with a pH of 6.5 or 4 for five days. Proteolytic activity was measured using 20 µL of sample (cell-free supernatant) and 80 µL of 7.5 µM bovine serum albumin (BSA) in 0.1 M citric acid/sodium phosphate, with a pH of 3.3, and incubated at 50 °C. Samples were incubated in the presence or absence of Pepstatin A (5, 10, and 15 μM) or PMSF (100 μM). Then, 20 µL were removed after 30 min and added to 180 µL of Coomassie PlusTM protein assay reagent (Pierce) in a microtiter plate. Absorbance was read at 590 nm, and activity (IU) was calculated relative to BSA degradation using a standard curve. One IU is defined as the degradation of 1 µmol of BSA min^−1^.mL^−1^. (**B**) Samples were incubated with collagen or hemoglobin for 30 min in the presence or absence of Pepstatin A. ** *p* ≤ 0.01, and *** *p* ≤ 0.001. The results shown are representative of three independent experiments.

**Table 1 jof-09-00375-t001:** Oligonucleotides used for real-time quantitative PCR analysis.

Gene		Sequence (5′→3)
*PbSAP **	Sense	GATGACTCTGAGGCTACCTTTG
	Anti Sense	ATCGAGATCAACCTCCCAGTA
*PbSAP2*	Sense	CCGTCTTCACCGCTCAATTA
	Anti Sense	CCACAGGGACATCAACCATATC
*PbYAP1*	Sense	GTCAACATGAGCGAGCTAGT
	Anti Sense	GAGATGCCGAAGATACAGGTT
*PbYAP2*	Sense	CCCGGTTATCTGTGAGAAAGTC
	Anti Sense	TGCGGATGACGTAGACAAAC
*α-TUB**	Sense	GTGGACCAGGTGATCGATGT
	Anti Sense	ACCCTGGAGGCAGTCACA
*18S **	Sense	CGGAGAGAGGGAGCCTGAGAA
	Anti Sense	GGGATTGGGTAATTTGCGC

* Reference [16].

**Table 2 jof-09-00375-t002:** Characteristics of the genes encoding aspartic proteases in *P. brasiliensis*.

Gene	Assession Number	Gene with Intron (bp)	Gene without Intron (bp)	Protein (aa)	Pep4 Similarity (%) *
*PbSAP*	PADG_00634	1460	1203	400	59
*PbSAP2 ***	PADG_03432	1571	1281	426	30
*PbYAP1 ***	PADG_12056#	1681	1485	495	27
*PbYAP2 ***	PADG_08282	1601	1446	481	29

***** Pep4 is aspartic peptidase in *S. cerevisiae.*
****** Named in this work.

**Table 3 jof-09-00375-t003:** Biochemical profile of aspartic proteases in *P. brasiliensis*.

Protein	Signal Peptide	Protein Molecular Weight (kDa)	Isoeletric Point (pI)	N-glycosylation Sites	PDB Similarity
PbSap	SSA-KV	43.8	5.36	N139, N339	Saccharopepsin—64%
PbSap2 *	VLG-AP	57	5.21	N164, N294	Aspergillopepsin—46%
PbYap1 *	VNG-LQ	51	5.03	N71, N131, N165, N341, N407	Candidapepsin—38%
PbYap2 *	VAA-FY	52.5	5.1	N172, N181, N416	Endopeptidase—30%

***** Named in this work.

**Table 4 jof-09-00375-t004:** Predicted functional partners of the aspartic proteases of *P. brasiliensis* detected using string network software.

Gene ID	Function	Interaction
PADG_06314	Carboxypeptidase Y homolog A; vacuolar carboxypeptidase	PbSap
PADG_07422	Uncharacterized protein; belongs to the peptidase S8 family	PbSap
PADG_05225	Orotidine 5′-phosphate decarboxylase; belongs to the OMP decarboxylase family	PbSap
PADG_11766	Alkaline phosphatase	PbSap
PADG_07910	Proteinase T [*B. dermatitidis* ATCC 18188]	PbSap
PADG_05608	Ras-like protein Rab7 [*B. dermatitidis* ATCC 18188]	PbSap
PADG_08109	Autophagy-related protein; belongs to the ATG8 family	PbSap
PADG_03562	Glucose-regulated protein [*B. dermatitidis* ATCC 18188]	PbSap
PADG_01553	Kexin [*B. dermatitidis* ATCC 18188]	PbSap, PbSap2, PbYap1 and PbYap2
PADG_04219	Glycosyl hydrolase [*B. dermatitidis* ATCC 18188]	PbSap
PADG_03290	Tripeptidyl peptidase SED3 [*B. dermatitidis* ATCC 18188]	PbSap2 and PbYap2
PADG_04488	Lipoate-protein ligase A [*B. dermatitidis* ATCC 18188]	PbYap2
PADG_08626	Uncharacterized protein	PbYap1
PADG_07283	Uncharacterized protein	PbYap2
PADG_11833	Hydrolase	PbYap2
PADG_02072	Uncharacterized protein	PbYap2
PADG_00011	Actin binding protein	PbYap2
PADG_00637	Arginase; belongs to the arginase family.	PbYap2
PADG_07781	Pyridoxine kinase	PbYap2
PADG_06133	Mating-type alpha-pheromone receptor PreB	PbYap2
PADG_08047	Serine/threonine-protein kinase ppk4	PbYap2

## Data Availability

The data presented in this study are available on request from the authors.

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
