# Peer review of "Characterization of Aspartic Proteases from Paracoccidioides brasiliensis and Their Role in Fungal Thermo-Dimorphism"

_jof, 2023, doi:10.3390/jof9030375_

Round 1

Reviewer 1 Report

The document entitled “Aspartic proteases from Paracoccidioides brasiliensis play a role in fungal-thermodimorphism and its virulence attributes” is submitted for special issue in the Journal of Fungi. The authors reveal 4 acid proteases that are present in the genome of this fungus. They then proceed to use the inhibitor pepstatin A to demonstrate the importance of acid proteases in the M to Y shift. They also demonstrate that these proteases are crucial for  yeast cells to survive exposure to a stressor. Overall, this paper does provide added information about proteases and the dimorphic shift, but it is deficient in multiple areas.

1.       The authors demonstrate that pepstatin A delay or inhibit the M to Y transition. Yet, we do not know whether this concentration of pepstatin actually inhibited the targeted proteases, and we do not know whether this concentration of pepstatin A altered the viability of the mycelia and that is why they could not transform into yeasts. The authors need to prove both of these issues to convince the reader of their claims. The evidence that aspartyl proteases influence transformation and/or virulence is indirect at best. More direct evidence is required.

2.       The optimal pH that they study is pH4. But where in the body is the pH that low. Even in the phagosomes the pH will not reach that low point. And in tissues, that pH is incompatible with life. Therefore, the studies of optimal expression at pH 4 have nothing to do with what might find in vivo.

3.       The authors state on line 508 that the fungus establishes infection in a low pH (but certainly not 4)  which is due to low oxygen. What evidence do they have that low oxygen modulates the pH? And if it does, shouldn’t some of these studies be conducted in low oxygen tensions?

Author Response

RESPOSNSE TO REVIEWER #1 COMMENTS

We would like to thank the reviewer #1 for careful and thorough reading of this manuscript and for the thoughtful comments and constructive suggestions, which help to improve the quality of this manuscript. Our response follows (the authors´ answers are in italic and blue):

Revisor #1

The document entitled “Aspartic proteases from Paracoccidioides brasiliensis play a role in fungal-thermodimorphism and its virulence attributes” is submitted for special issue in the Journal of Fungi. The authors reveal 4 acid proteases that are present in the genome of this fungus. They then proceed to use the inhibitor pepstatin A to demonstrate the importance of acid proteases in the M to Y shift. They also demonstrate that these proteases are crucial for yeast cells to survive exposure to a stressor. Overall, this paper does provide added information about proteases and the dimorphic shift, but it is deficient in multiple areas.

  1. The authors demonstrate that pepstatin A delay or inhibit the M to Y transition. Yet, we do not know whether this concentration of pepstatin actually inhibited the targeted proteases, and we do not know whether this concentration of pepstatin A altered the viability of the mycelia and that is why they could not transform into yeasts. The authors need to prove both of these issues to convince the reader of their claims. The evidence that aspartyl proteases influence transformation and/or virulence is indirect at best. More direct evidence is required.

Response: We appreciate the reviewer's comment to improve our article. First, it is necessary to explain Pepstatin A and its use in fungal models. Pepstatin A is a classic inhibitor of aspartic proteases (reviewed in Braga-Silva and Santos, 2011) and widely used in assays to evaluate its activity (both in vitro and in vivo assay) (Zotter et al., 1990; Naglik et al., 2008; Dalle et al., 2010; Pericolini et al., 2015; Purushothaman et al., 2019; Charkraborty et al., 2023). In the present study, we used a concentration of 2.5 μM of Pepstatin A to assess whether the inhibition of aspartic proteases would play a role in the dimorphic change of the fungus. This concentration is not toxic to the fungus in spot assay. We tested several inhibitor concentrations (0.01 to 10 μM), and none showed cell death (Figure A). Furthermore, we incubated the fungus at a concentration of 2.5 and 5 μM and observed viability for 24, 48, and 72 h and did not observe differences in viability loss (Figure B). Different studies used Pepstatin A in an in vivo assay (application in yeasts) and did not observe a loss of viability up to a concentration of 50 μM (Dalle et al., 2010). In this new manuscript version, we have included Figures A and B as Supplementary Figure 1.

Figure A. Evaluation of the cytotoxicity of Pepstatin A in P. brasiliensis. Yeast cells (1 x 104) were cultured in microplates containing RPMI medium and treated with Pepstatin A (0.018 - 10 µM), Methanol (vehicle) or Itraconazole (0.01 - 10 µM) (Positive Control). The culture was incubated for 7 days at 37°C and after that period 10 μL of each point was applied in YPDmod medium and incubated again for 5 days at 37°C. Representative data from three independent experiments.

Figure B. Viability profile of P. brasiliensis cultured in the presence of Pepstatin A. Yeast cells (1 x104) were plated in mYPD medium in the presence or absence of pepstatin A (2.5 or 5 μM) at 37°C for 24, 48 and 72 h. Inhibitor was added daily. The fungal growth was determined by counting in a Neubauer chamber. Representative data from three independent experiments.

- Braga-Silva LA, Santos AL. Aspartic protease inhibitors as potential anti-Candida albicans drugs: impacts on fungal biology, virulence and pathogenesis. Curr Med Chem. 2011; 18(16):2401-19. doi: 10.2174/092986711795843182.

- Naglik JR, et al. Quantitative expression of the Candida albicans secreted aspartyl proteinase gene family in human oral and vaginal candidiasis. Microbiology. 2008; 154(Pt 11):3266-3280. doi: 10.1099/mic.0.2008/022293-0.

- Dalle F et al., Cellular interactions of Candida albicans with human oral epithelial cells and enterocytes. Cell Microbiol. 2010; 12(2):248-71. doi: 10.1111/j.1462-5822.2009.01394.x.

- Chakraborty S. et al., Expanding the therapeutic options for Candida infections using novel inhibitors of secreted aspartyl proteases. Drug Dev Res. 2023; 84(1):96-109. doi: 10.1002/ddr.22015.

- Pericolini E et al. Secretory Aspartyl Proteinases Cause Vaginitis and Can Mediate Vaginitis Caused by Candida albicans in Mice. mBio. 2015; 6(3):e00724. doi: 10.1128/mBio.00724-15.

- Purushothaman K et al. Aspartic protease from Aspergillus niger: Molecular characterization and interaction with pepstatin A. Int J Biol Macromol. 2019. 15;139:199-212. doi: 10.1016/j.ijbiomac.2019.07.133

- Zotter C. et al. Effect of pepstatin A on Candida albicans infection in the mouse. Dermatol Monatsschr. 1990;176(2-3):189-98.

  1. The optimal pH that they study is pH4. But where in the body is the pH that low. Even in the phagosomes the pH will not reach that low point. And in tissues, that pH is incompatible with life. Therefore, the studies of optimal expression at pH 4 have nothing to do with what might find in vivo.

Response: Our group has studied the influence of pH on the secretion of some virulence factors. A previous study of our group demonstrated that PbSap is highly secreted at pH 4, compared to pH 6.5 (control) (Castilho et al., 2018). In the present study, we verified the role of aspartic proteases, which are only active at acidic pH. Most of the aspartic proteases display best enzyme activity at low pH (pH 3 to 4) (Rao et al., 1998; review in Hofer et al., 2020). Other studies have also used pH 4 to assess the role of aspartic proteases in fungal biology and pathogenesis (Konno et al., 2006; Sherrington et al., 2017; Sriranganadane et al., 2010). In addition, site-specific acidic pH studies have been performed with other pathogens, for example: in the stomach, Salmonella enterica or Listeria monocutogenes survive and cause disease at pH 1-2; the vaginal epithelium at pH 4 and in the colon pH 5.7 allow Candida albicans infection (review in Luand et al., 2020). Although there is still no evidence of colonization of these tissues by P. brasiliensis, these studies show that certain anatomical locations have low pH. Another point to be considered is the adaptation and survival within phagolysosomes. There are still controversies in the literature about the pH value in this cell compartment, but the pH of lysosomes is well characterized, which varies between pH 4.5 – 5.0 (review in Zeng et al., 2020). In macrophages, a pH of 4.5 has been measured, while in Dictyostelium discoideum a lower phagosomal pH of 3.5–4.5 has been observed (Aubry et al., 1993; Gopaldass et al., 2012; Yates et al., 2005; Marchetti et al. al., 2009; Sattler et al., 2013). These studies suggest that P. brasiliensis may be subjected to these pH values after phagocytosis. However, we agree with the reviewer that the pH values in physiological conditions in which the fungus would have contact are still debatable. During the immune response within phagocytic cells, although the pH value is still somewhat debatable, the fungus can come into contact with low pH. Thus, we changed some points of the manuscript so that this correlation does not cause confusion in the reader and in the community. We hope the reviewer agrees with the changes.

- Castilho, DG. et al. Secreted aspartyl proteinase (PbSap) contributes to the virulence of Paracoccidioides brasiliensis infection. PLOS Neglected Tropical Diseases 2018, 12, e0006806, doi:10.1371/journal.pntd.0006806.

- Rao MB. Et al. Molecular and biotechnological aspects of microbial proteases. Microbiology and Molecular Biology Reviews. 1998; 62(3):597–635.

- Hofer F. et al. Catalytic Site pKa Values of Aspartic, Cysteine, and Serine Proteases: Constant pH MD Simulations. J Chem Inf Model. 2020; 60(6): 3030–3042. doi: 10.1021/acs.jcim.0c00190

- Konno N. et al. Mechanism of Candida albicans transformation in response to changes of pH. Biol Pharm Bull. 2006; 29(5):923-6. doi: 10.1248/bpb.29.923.

- Sherrington SL. et al. Adaptation of Candida albicans to environmental pH induces cell wall remodelling and enhances innate immune recognition. PLoS Pathog. 2017; 13(5): e1006403. doi: 10.1371/journal.ppat.1006403

- Sriranganadane D. et al. Aspergillus protein degradation pathways with different secreted protease sets at neutral and acidic pH. J Proteome Res. 2010; 9(7):3511-9. doi: 10.1021/pr901202z.

- Lund PA. et al. Understanding How Microorganisms Respond to Acid pH Is Central to Their Control and Successful Exploitation. Front Microbiol. 2020; 11:556140. doi: 10.3389/fmicb.2020.556140.

- Zeng J. et al. Modulating lysosomal pH: a molecular and nanoscale materials design perspective. J Life Sci (Westlake Village). 2020; 2(4): 25–37. doi: 10.36069/jols/20201204

- Aubry L, et al. Kinetics of endosomal pH evolution in Dictyostelium discoideum amoebae. Study by fluorescence spectroscopy. J Cell Sci. 1993; 105: 861–866

- Gopaldass N. et al. Dynamin A, myosin IB and Abp1 couple phagosome maturation to F-actin binding. Traffic. 2012; 13:120–130.

- Yates RM, Hermetter A, Russell DG. The kinetics of phagosome maturation as a function of phagosome/lysosome fusion and acquisition of hydrolytic activity. Traffic. 2005; 6:413–420.

- Marchetti A, Lelong E, Cosson P. A measure of endosomal pH by fl ow cytometry in Dictyostelium . BMC Res Notes. 2009; 2:7.

- Sattler, N., Monroy, R., and Soldati, T. Quantitative Analysis of Phagocytosis and Phagosome Maturation. Methods Mol. Biol. 2013; 983, 383–402. doi: 10.1007/978-1-62703-302-2_21.

  1. The authors state on line 508 that the fungus establishes infection in a low pH (but certainly not 4) which is due to low oxygen. What evidence do they have that low oxygen modulates the pH? And if it does, shouldn’t some of these studies be conducted in low oxygen tensions?

Response: We agree with the reviewer that this statement is inaccurate in the main text. Actually, the most likely mechanism is that the fungi are submitted to low concentration of oxygen in the host tissues, and to survive their can carry out fermentation. As a result of fermentation by-products, the pH of the environment can be altered and become acidified. In the work by Han et al., (2018) the authors discuss the aspect of the modulation of aspartic proteases being expressed in conditions of hypoxia and the decrease in pH in response to the proliferation of the pathogen. In this revised version of the manuscript, we removed the sentence on line 508 because we also understand that our data are not related to tissue hypoxia, making this information unnecessary for a complete understanding of the work. We hope that the reviewer agrees with these changes.

Han Z et al. Growth and protease secretion of Scedosporium aurantiacum under conditions of hypoxia. Microbiol Res. 2018; 216:23-29. doi: 10.1016/j.micres.2018.08.003.

Again, we appreciate all your insightful comments. Thank you for taking the time to help us improve the paper.

Reviewer 2 Report

The paper presents an extensive and multidirectional study of proteases produced by P. brasiliensis. Manuscript is well organized, below are my small comments:

2.1. Was the research done with animals overseen by the appropriate bioethics committee?

Table 1. Oligonucleotides - Were the listed starters designed by the authors of the paper? if not please provide relevant references

Spelling errors:

289 "Caltalytic sites"

308 " iden-tified"

316 " sug-gesting"

Author Response

RESPONSE TO REVIEWER #2 COMMENTS

We would like to thank the reviewer #2 for careful and thorough reading of this manuscript and for the thoughtful comments and constructive suggestions, which help to improve the quality of this manuscript. Our response follows (the authors´ answers are in italic and blue):

Revisor #2

The paper presents an extensive and multidirectional study of proteases produced by P. brasiliensis. Manuscript is well organized, below are my small comments:

2.1. Was the research done with animals overseen by the appropriate bioethics committee?

Response: In this work we do not use animals, but only the fungus isolated from the animal. This procedure is always performed by our group to recover the fungus virulence. In this revised manuscript version, we have included the animal ethics committee number regarding this fungal virulence recovery procedure (CEUA 8888301117). We apologize for not including this information earlier.

Table 1. Oligonucleotides - Were the listed starters designed by the authors of the paper? if not please provide relevant references.

Response: Yes, oligonucleotides to analysis the expression of PbSAP2, PbYAP1 and PbYAP2 were designed for the present work. On the other hand, the PbSAP, α-TUB and 18S oligonucleotides were validated in another paper by our group (Castilho et al., 2018). This information was included in the revised manuscript version.

Castilho, DG. et al. Secreted aspartyl proteinase (PbSap) contributes to the virulence of Paracoccidioides brasiliensis infection. PLOS Neglected Tropical Diseases 2018, 12, e0006806, doi:10.1371/journal.pntd.0006806.

Spelling errors:

289 "Caltalytic sites"

308 " iden-tified"

316 " sug-gesting"

Response: We thank the reviewer for this observation. These mistakes were corrected in the revised version of the manuscript.

Again, we appreciate all your insightful comments. Thank you for taking the time to help us improve the paper.

Reviewer 3 Report

The manuscript “Aspartic proteases from Paracoccidioides brasiliensis play a

role in fungal thermo-dimorphism and its virulence attributes” bring to light the effect of pepstatin a treatment on the biology of this pathogen. Although the manuscript significantly contributes to the field by bringing new putative aspects of aspartic proteases into the model, the conclusions drawn are not supported by the provided data. Thus, the manuscript must be adjusted before being accepted in this journal.

Major:

The title must be changed in order to be representative of the obtained data. For example, there is no evidence that aspartic protease affects the virulence of the fungus.

Lines 20 and 21: this cannot be assumed since it was used a protease inhibitor and not a functional analysis.

Lines 26-28: it was not demonstrated any relation between the protease activity and pathogenesis

Lines: 234-6: the authors should clarify how they identify the new protease gene. They should provide figures to explain.

Table 2: please use more appropriate names in the column headings.

Lines 246-58: the authors should avoid repeating all the data already present in the tables. They should connect the features to the function.

Lines 299-317: The rationale for STRING-based analysis needs to be clarified. Also, why the interaction network appears in two groups? What is the meaning of the line colors? It is necessary to be cautious in assuming that any interaction in STRING is biologically significant. The lack of a rationale for the analysis reflects the poor discussion of these results.

Lines 327-8: please, show the data and the reason for the usage of 2.5uM dose.

Line 341: The assay does not show that the aspartic protease regulates the phenomenon. It only showed that treatment with pepstatin affects M to Y transition.

Item 3.3: What is the rationale for using BSA in this assay?

Lines 396-401: basic/neutral pH conditions should be used as controls.

Lines 400-1: conclusion not supported by the data.

The description of data presented in figure 4B is very confusing. Why the YAP1 was not considered as regulated?

Lines 417-9: “expression considered high” please clarify

Lines 413-4: what is the biological meaning of this observation if glucose is considered a carbon source and peptone a C and N source?

Lines 419-21: broad conclusion. Please narrow the conclusions based on the results.

Lines 480-1: The conclusion is not supported by the data.

Lines 519: the inhibition was partial.

Line 535: partial inhibition.

Lines 575-96: background section paragraphs.

608-20: literature review

634-5: it is not supported by the results.

638: the data does not show full inhibition of protease activity by pepstatin

Lines 659-65: meaningless

667-73: the conclusion should be more narrowed and specifically connected to the obtained results.

The discussion section is not integrative and interpretative. The section is filled with results descriptions and background information. The authors must integrate and link their data with the available information. Please, adjust it accordingly.

Minor:

Line 39: public? Please correct this term.

Lines 65-72: The functions and localization of the cellular aspartic proteases need to be clarified. Mainly for the vacuolar ones.

Lines: 85-6: Please clarify: “the aspartyl proteases in P. brasiliensis have not yet been provided and their functions have not yet been described”

Line 257: are the similarities based on sequence or structure?

Line 308: “iden-tified”? Please correct this kind of typo throughout the manuscript.

Lines 313-4: confuse. Please clarify.

The manuscript must be subjected to language editing to get English improvement.

Author Response

RESPONSE TO REVIEWER #3 COMMENTS

We would like to thank the reviewer #3 for careful and thorough reading of this manuscript and for the thoughtful comments and constructive suggestions, which help to improve the quality of this manuscript. Our response follows (the authors´ answers are in italic and blue):

Revisor #3

The manuscript “Aspartic proteases from Paracoccidioides brasiliensis play a role in fungal thermo-dimorphism and its virulence attributes” bring to light the effect of pepstatin a treatment on the biology of this pathogen. Although the manuscript significantly contributes to the field by bringing new putative aspects of aspartic proteases into the model, the conclusions drawn are not supported by the provided data. Thus, the manuscript must be adjusted before being accepted in this journal.

Major:

The title must be changed in order to be representative of the obtained data. For example, there is no evidence that aspartic protease affects the virulence of the fungus.

Response: We thanks the reviewer for this suggestion. Thus, in this revised manuscript version we have changed the title to: “Characterization of aspartic proteases from Paracoccidioides brasiliensis and their role in fungal thermo-dimorphism”. We hope the reviewer agrees with the change.

Lines 20 and 21: this cannot be assumed since it was used a protease inhibitor and not a functional analysis.

Response: Pepstatin A is a classic aspartic protease inhibitor (Braga and Santos, 2011). Thus, we assumed that the delay in fungal dimorphism would be related to aspartic proteases inhibition. However, we agree that additional experiments could confirm this hypothesis. Thus, we modify the sentence to: “We demonstrated that Pepstatin A can inhibit dimorphic switching (mycelium → yeast) in P. brasiliensis.” (lines 20-21). In addition, we have included a new supplementary figure (Supplemenary Figure 1) that shows the fungal viability after Pepstain A treatment, demonstrating that the delay in the transition form is not due to a possible toxic effect of the inhibitor. We hope the reviewer agrees with the change.

- Braga-Silva LA, Santos AL. Aspartic protease inhibitors as potential anti-Candida albicans drugs: impacts on fungal biology, virulence and pathogenesis. Curr Med Chem. 2011; 18(16):2401-19. doi: 10.2174/092986711795843182.

Lines 26-28: it was not demonstrated any relation between the protease activity and pathogenesis

Response: We agree with the reviewer. In this revised manuscript version, we have modified the sentence to: “These data suggest that aspartyl proteases are modulated by environmental conditions and during fungal thermo-dimorphism. Thus, this work brings new possibilities for studying the role of aspartyl proteases in the host-pathogen relationship and P. brasiliensis biology.” (lines 26-28). We hope that the reviewer agrees with these changes.

Lines: 234-6: the authors should clarify how they identify the new protease gene. They should provide figures to explain.

Response: The description of aspartic protease genes identification is described in section Material and Methods item 2.7 (lines 184-189).

Table 2: please use more appropriate names in the column headings.

Response: In this revised manuscript version we have modified the column headings.

Lines 246-58: the authors should avoid repeating all the data already present in the tables. They should connect the features to the function.

Response: We appreciate the reviewer's suggestion. In this revised manuscript version, we have modified the sentence so as not to repeat information presented in the table. I hope the reviewer agrees with the change (lines 246-256).

Lines 299-317: The rationale for STRING-based analysis needs to be clarified. Also, why the interaction network appears in two groups? What is the meaning of the line colors? It is necessary to be cautious in assuming that any interaction in STRING is biologically significant. The lack of a rationale for the analysis reflects the poor discussion of these results.

Response: In this study we used STRING analyzes to identify possible protein interactions with aspartyl proteases. These interactions could help in the development of other assays to better understand the role of aspartyl proteases in fungal biology. From these STRING studies, we identified the interaction of PbSap with ATG8, an important component of the autophagy pathway (Varga et al., 2022). Ribeiro et al. (2018), demonstrated that autophagy is an important process for occurring the thermo-dimorphism of P. brasiliensis (Ribeiro et al., 2018). Furthermore, in other fungi, the role of aspartic proteases in autophagy is better demonstrated (Cortez-Sánchez et al., 2018). Due to this result we decided to investigate whether aspartic proteases could be regulated during M-Y thermo-dimorphism. As shown in Figure 4A all aspartic protease genes were up-regulated during term-dimorphism. Consequently, we used Pepstatin A (an important aspartyl protease inhibitor) in the dimorphic transition assay and we observed a delay in morphological change (Figure 2). These results suggest that aspartic proteases can are involved in transition form induced by temperature. Our group is now investigating the mechanisms involved in this transition and how aspartyl proteases participate in this process.

Regarding the result of the two groups of STRING analysis, this occurred due to a notation problem in the Pb18 genome of P. brasiliensis. The sequence PADG_12056 (which we named PbYap1) was found in the Fungi DB database with the notation of "C Subunit Replication Fact". However, after studying the sequence displayed by the bank, we detected an error and, when comparing it with the same sequence from another strain of Paracoccidioides (Pb03), we identified that the PADG_12056 protein is the junction of two genes that codify two proteins, being identified in the Pb3 genome as PABG_06949 and PABG_06947. Thus, we show that PADG_12056 consists of two predictor genes joined together. In the present study, all PADG_12056 sequence analyzes were performed between amino acids 360 and 846, corresponding to the PbYAP1 sequence.

During the STRING analyses, it was not possible to use the FASTA sequence of PbYAP1 (PADG_12056) because, as previously mentioned, the sequence was truncated with two genes, so we used the FASTA sequence of the Pb01 strain (PAAG_03619), since it was not possible to use the sequence FASTA from Pb03. For this reason, the software formed two distinct groups. As this is an analysis that was not carried out with the Pb18 genome, due to the problems mentioned above, we decided to place this figure (Figure 1B, direct panel) as a supplementary figure.

The meaning of the colored lines has been added in the Figure 1 of the revised manuscript.

- Cortez-Sánchez JL. et al. Activity and expression of Candida glabrata vacuolar proteases in autophagy-like conditions. FEMS Yeast Res. 2018; 18(2). doi: 10.1093/femsyr/foy006.

- Varga VB. et al. The evolutionary and functional divergence of the Atg8 autophagy protein superfamily. Biol Futur. 2022; 73(4):375-384. doi: 10.1007/s42977-022-00123-6.

- Ribeiro GF. et al. Autophagy in Paracoccidioides brasiliensis under normal mycelia to yeast transition and under selective nutrient deprivation. PLoS One. 2018; 13(8):e0202529. doi: 10.1371/journal.pone.0202529.

Lines 327-8: please, show the data and the reason for the usage of 2.5 uM dose.

Response: We used the concentration of 2.5 μM Pepstatin A, because this dose is in the concentration range capable of inhibiting the enzymatic activity (Aoki et al., 2012; Purushothaman et al., 2019). This concentration is not toxic to the fungus in spot assay. We tested several inhibitor concentrations (0.018 to 10 μM) and none of them showed cell death (Figure A). Furthermore, we incubated the fungus at a concentration of 2.5 and 5 μM and observed viability for 24, 48 and 72h and did not observe differences in viability loss (Figure B).

Figure A. Evaluation of the cytotoxicity of Pepstatin A in P. brasiliensis. Yeast cells (1 x 104) were cultured in microplates containing RPMI medium and treated with Pepstatin A (0.018 - 10 µM), Methanol (vehicle) or Itraconazole (0.01 - 10 µM) (Positive Control). The culture was incubated for 7 days at 37°C and after that period 10 μL of each point was applied in YPDmod medium and incubated again for 5 days at 37°C. Representative data from three independent experiments.

Figure B. Viability profile of P. brasiliensis cultured in the presence of Pepstatin A. Yeast cells (1 x104) were plated in mYPD medium in the presence or absence of pepstatin A (2.5 or 5 μM) at 37°C for 24, 48 and 72 h. Inhibitor was added daily. The fungal growth was determined by counting in a Neubauer chamber. Representative data from three independent experiments.

- Aoki W. et al. Design of a novel antimicrobial peptide activated by virulent proteases. Chem Biol Drug Des. 2012; 80(5):725-33. doi: 10.1111/cbdd.12012.

- Purushothaman K et al. Aspartic protease from Aspergillus niger: Molecular characterization and interaction with pepstatin A. Int J Biol Macromol. 2019. 15;139:199-212. doi: 10.1016/j.ijbiomac.2019.07.133

Line 341: The assay does not show that the aspartic protease regulates the phenomenon. It only showed that treatment with pepstatin affects M to Y transition.

Response: Pepstatin A is a classic specific inhibitor of the catalytic site of aspartic proteases (reviewed in Braga-Silva and Santos, 2011) and widely used in assays to evaluate its activity (both in vitro and in vivo assay) (Zotter et al., 1990; Naglik et al., 2008; Dalle et al., 2010; Pericolini et al., 2015; Purushothaman et al., 2019; Charkraborty et al., 2023). In the present study, we used Pepstatin A (2.5 μM) to assess whether the inhibition of aspartic proteases would play a role in the dimorphic change of the fungus. However, we agree that additional experiments could confirm this hypothesis. So, we modify the sentence to: “These data demonstrate that Pepstatin A regulates dimorphic switching (M→Y) in P. brasiliensis.” (lines 313-314). We hope the reviewer agrees with the change.

- Braga-Silva LA, Santos AL. Aspartic protease inhibitors as potential anti-Candida albicans drugs: impacts on fungal biology, virulence and pathogenesis. Curr Med Chem. 2011; 18(16):2401-19. doi: 10.2174/092986711795843182.

- Naglik JR, et al. Quantitative expression of the Candida albicans secreted aspartyl proteinase gene family in human oral and vaginal candidiasis. Microbiology. 2008; 154(Pt 11):3266-3280. doi: 10.1099/mic.0.2008/022293-0.

- Dalle F et al., Cellular interactions of Candida albicans with human oral epithelial cells and enterocytes. Cell Microbiol. 2010; 12(2):248-71. doi: 10.1111/j.1462-5822.2009.01394.x.

- Chakraborty S. et al., Expanding the therapeutic options for Candida infections using novel inhibitors of secreted aspartyl proteases. Drug Dev Res. 2023; 84(1):96-109. doi: 10.1002/ddr.22015.

- Pericolini E et al. Secretory Aspartyl Proteinases Cause Vaginitis and Can Mediate Vaginitis Caused by Candida albicans in Mice. mBio. 2015; 6(3):e00724. doi: 10.1128/mBio.00724-15.

- Purushothaman K et al. Aspartic protease from Aspergillus niger: Molecular characterization and interaction with pepstatin A. Int J Biol Macromol. 2019. 15;139:199-212. doi: 10.1016/j.ijbiomac.2019.07.133

- Zotter C. et al. Effect of pepstatin A on Candida albicans infection in the mouse. Dermatol Monatsschr. 1990;176(2-3):189-98.

Item 3.3: What is the rationale for using BSA in this assay?

Response: Previously, our group demonstrated that the supplementation with BSA (organic nitrogen source) induces PbSAP production in P. brasiliensis (Castilho et al., 2018). Thus, we wanted to verify whether the addition of this organic nitrogen source could modulate fungal dimorphism. As observed in Figure 2, no difference was verified in the dimorphic transition. This information was included in the revised manuscript.

Castilho, DG. et al. Secreted aspartyl proteinase (PbSap) contributes to the virulence of Paracoccidioides brasiliensis infection. PLOS Neglected Tropical Diseases 2018, 12, e0006806, doi:10.1371/journal.pntd.0006806.

Lines 396-401: basic/neutral pH conditions should be used as controls.

Response: It is important to inform that pH 6.5 is the control condition in this assay. P. brasiliensis normally grows in YPD medium at pH 6.5. However, we agreed that the sentence was confusing, so we rewrote the sentence (line 400). We hope the reviewer agrees with the change.

Lines 400-1: conclusion not supported by the data.

Response: We change the conclusion sentence to: “These data indicate that these genes are responsive to pH 4 in P. brasiliensis.” (lines 403-404).

The description of data presented in figure 4B is very confusing. Why the YAP1 was not considered as regulated?

Response: We agree with the reviewer that some sentences in the paragraph were confusing. Thus, we have modified this paragraph. I hope the reviewer agrees with the change.

It is also important to inform that we consider the expression of PbYAP1 to be positively regulated, according to the text: “PbYAP1 gene expression was significantly increased in both conditions, in the absence of nutrients (3.3-fold) and the presence of peptone (7-fold) (Figure 4B).” (lines 411-425). We said that in the presence of the carbon source (dextrose) the induction of PbYAP1 did not occur, as shown in figure 4B.

Lines 417-9: “expression considered high” please clarify.

Response: We agree that the sentence was confusing, so in this revised manuscript version we have corrected the sentence (line 420).

Lines 413-4: what is the biological meaning of this observation if glucose is considered a carbon source and peptone a C and N source?

Response: In the present study, we evaluated whether the presence or absence of an organic carbon (dextrose) or organic nitrogen (peptone) source could regulate the aspartic proteases expression. Several works have shown peptone as an organic nitrogen source (Li et al., 2000; Delgado-Jarana et al., 2000; Lario et al., 2020; Sun et al., 2021, Souza et al., 2017; Wu et al., 2019). In addition, the peptone have been involved in the production, secretion and activation of proteases (Li et al., 2000; Delgado-Jarana et al., 2000; Lario et al., 2020; Sun et al., 2021, Souza et al., 2017; Wu et al., 2019). We have not found literature considering peptone as a carbon source.

- Li Y. et al. Effect of nitrogen source and nitrogen concentration on the production of pyruvate by Torulopsis glabrata. J Biotechnol. 2000; 28;81(1):27-34. doi: 10.1016/s0168-1656(00)00273-x.

- Delgado-Jarana J. et al. Overproduction of beta-1,6-glucanase in Trichoderma harzianum is controlled by extracellular acidic proteases and pH. Biochim Biophys Acta. 2000; 1481(2):289-96. doi: 10.1016/s0167-4838(00)00172-2.

- Lario LD. et al. Optimization of protease production and sequence analysis of the purified enzyme from the cold adapted yeast Rhodotorula mucilaginosa CBMAI 1528. Biotechnol Rep (Amst). 2020; 28:e00546. doi: 10.1016/j.btre.2020.e00546.

- Sun Y. et al. Extracellular protease production regulated by nitrogen and carbon sources in Trichoderma reesei. Journal of Basic Microbiology. 2021; 61:122-132. doi: 10.1002/jobm.202000566.

- Souza PM. et al. Production, purification and characterization of an aspartic protease from Aspergillus foetidus. Food Chem Toxicol. 2017; 109(Pt 2):1103-1110. doi: 10.1016/j.fct.2017.03.055.

- Wu R. et al. Cost-effective fibrinolytic enzyme production by Bacillus subtilis WR350 using medium supplemented with corn steep powder and sucrose. Sci Rep. 2019; 9: 6824. doi: 10.1038/s41598-019-43371-8

Lines 419-21: broad conclusion. Please narrow the conclusions based on the results.

Response: We have modified the sentence as requested. We hope the reviewer agrees with the change (lines 422 – 424).

Lines 480-1: The conclusion is not supported by the data.

Response: We modified the sentence as suggested (lines 479-480).

Lines 519: the inhibition was partial.

Response: We modified the sentence as suggested (line 517).

Line 535: partial inhibition.

Response: We modified the sentence as suggested (line 534).

Lines 575-96: background section paragraphs. 608-20: literature review

Response: We did not find what action should be performed to improve the manuscript. We improved the discussion section as requested below.

634-5: it is not supported by the results.

Response: We agree with the reviewer that the sentence is not consistent with the results obtained. Thus, we removed this statement from the paragraph.

638: the data does not show full inhibition of protease activity by pepstatin

Response: We have modified the sentence. We detected an error in Figure 6, we used in these assays 5 µM of Pepstatin A (as stated in the main text of the manuscript) and not 10 µM as stated in the figure and legend. This has been corrected in the new Figure 6. In addition, we tested higher inhibitor concentrations (10 and 15 µM) and observed inhibition of activity in a dose-response manner. These new data were included in Figure 6 of the revised manuscript version.

Lines 659-65: meaningless

Response: We have modified this paragraph to make it more specific and reflect the observed result. We hope the reviewer agrees with this change.

667-73: the conclusion should be more narrowed and specifically connected to the obtained results.

Response: We agree with the reviewer. In this revised manuscript version, we modified the conclusion section making it more specific in relation to the results obtained (lines 666-674). We hope the reviewer agrees with the changes.

The discussion section is not integrative and interpretative. The section is filled with results descriptions and background information. The authors must integrate and link their data with the available information. Please, adjust it accordingly.

Response: We agree with this comment. Therefore, we modified the discussion section to make it more integrative and interactive. We hope the reviewer agrees with the changes.

Minor:

Line 39: public? Please correct this term.

Response: The correct is “public health”. This has been corrected in the revised manuscript version.

Lines 65-72: The functions and localization of the cellular aspartic proteases need to be clarified. Mainly for the vacuolar ones.

Response: The requested information can be found in the main text of the manuscript.

Lines: 85-6: Please clarify: “the aspartyl proteases in P. brasiliensis have not yet been provided and their functions have not yet been described”

Response: We agree with the reviewer. In this revised manuscript version, we modified the sentence (lines 84-85). We hope the reviewer agrees with the changes.

Line 257: are the similarities based on sequence or structure?

Response: Similarity was based on sequence.

Line 308: “iden-tified”? Please correct this kind of typo throughout the manuscript.

Response: These mistakes were corrected.

Lines 313-4: confuse. Please clarify.

Response: We have modified this sentence. We hope the sentence was clear and objective.

The manuscript must be subjected to language editing to get English improvement.

Response: We have extensively edited the text to correct manuscript problems.

Again, we appreciate all your insightful comments. Thank you for taking the time to help us improve the paper.

Round 2

Reviewer 1 Report

The authors have responded to my queries. However, questions remain.

1. The authors refer to past literature on the effect of pepstatin A and fungi. But that literature does not mean that in this fungus at those concentrations that the protease activity is altered. Nor, does it exclude off-target effects of which pepstatin A has many in mammalian cells. I think the least the authors should do is prove that the protease activity is dampened by pepstatin A.

2. The authors argue about pH. They refer to the intestine and the vaginal vault. But those are luminal and indeed they are low. But the phagosome of a macrophage infected with a pathogen may drop to 5.5 as a mechanism to kill the fungus. However, many intracellular microbes avoid this pH in order to survive. One cannot correlate the pH of a luminal environment with that of a phagosome. That is a false comparison. The authors need to provide information regarding the pH of the phagosome harboring Paracoccidioides and then use that pH. Just because they have used pH 4 in the past does not mean that it is biologically relevant to the in vivo situation. 

Author Response

REQUESTED REVISIONS - JoF

Authors’ answers

Rafael Souza da Silva, Wilson Dias Segura, Reinaldo Souza Oliveira, Patricia Xander, Wagner Luiz Batista

RESPOSNSE TO REVIEWER #1 COMMENTS

We would like to thank the reviewer #1 for careful and thorough reading of this manuscript and for the thoughtful comments and constructive suggestions, which help to improve the quality of this manuscript. Our response follows (the authors´ answers are in italic and blue):

Revisor #1

The authors have responded to my queries. However, questions remain.

  1. The authors refer to past literature on the effect of pepstatin A and fungi. But that literature does not mean that in this fungus at those concentrations that the protease activity is altered. Nor, does it exclude off-target effects of which pepstatin A has many in mammalian cells. I think the least the authors should do is prove that the protease activity is dampened by pepstatin A.

Response: We thank the reviewer for his concern about improving our work. In this review of the article, we tested the concentrations of Pepstatin A used in work to assess whether they could inhibit the aspartic protease activity in P. brasiliensis. For that, the protein extract of the fungus cultivated under control and pH 4 conditions was used to evaluate the activity of aspartic proteases in the presence and absence of pepstatin A (1; 2.5, 5 μM). Figure A shows that the aspartic protease activity was higher at pH 4 compared to the control condition. Furthermore, in both conditions, there was a dose-dependent reduction in proteolytic activity in the presence of pepstatin A. On the other hand, no significant reduction in proteolytic activity was observed when cultured at pH 6.5 or 4 in the presence of PMSF inhibitor. This result was inserted in the revised version of the manuscript as Supplementary Figure 2 (lines 299-305). We hope to have answered the reviewer's question.

 Please, see figure the attachment. 

Figure A. Proteolytic activity profile of acid proteases from P. brasiliensis. Yeast cells were grown in YPDm with pH 6.5 or 4 for five days. Proteolytic activity was measured using 3 µg P. brasiliensis protein extract and 80 µL of 7.5 µM bovine serum albumin (BSA) in 0.1 M citric acid/sodium phosphate, pH 3.3, and incubated at 50°C. Samples were incubated in the presence or absence of Pepstatin A (1, 2.5 and 5 μM) or PMSF (100 μM). Then, 20 µL was removed after 30 min and added to 180 µL of Coomassie PlusTM protein assay reagent (Pierce) in a microtiter plate. Absorbance was read at 590 nm, and activity (IU) was calculated relative to BSA degradation using a standard curve. One IU is defined as degradation of 1 µmol of BSA min-1.ml-1. *p≤0.05 and **p≤0.01. The results shown are representative of two independent experiments.

  1. The authors argue about pH. They refer to the intestine and the vaginal vault. But those are luminal and indeed they are low. But the phagosome of a macrophage infected with a pathogen may drop to 5.5 as a mechanism to kill the fungus. However, many intracellular microbes avoid this pH in order to survive. One cannot correlate the pH of a luminal environment with that of a phagosome. That is a false comparison. The authors need to provide information regarding the pH of the phagosome harboring Paracoccidioides and then use that pH. Just because they have used pH 4 in the past does not mean that it is biologically relevant to the in vivo situation.

Response. We appreciate the reviewer's comment. In the first review, we discuss the different environments with acidic pH that fungi can find in the host. We only detail that because the reviewer asked about sites with low pH in the body. However, we agree with the reviewer that these sites are specific, so we have removed them from the manuscript. In addition, as suggested by the reviewer, we removed any mention of the relationship between our data and the conditions of survival in the phagolysosome. It is important to clarify that the choice of using pH 4 was to better understand P. brasiliensis biology with a focus on the activity and secretion of aspartic proteases. Different studies carried out this approach to study these enzymes with other fungi, as evidenced by the literature. Our research group has also dedicated itself to studying "biologically relevant" models, although this expression is controversial. Science must be open and inclusive to basic studies since these studies support the evidence and the deepening of "biologically relevant" techniques and models. There are countless examples of this, such as the study with bacteriophages, which today has shown therapeutic application, identification, and analysis of extremophilic bacteria, which allowed the development of PCR, not to mention the studies in physics and chemistry that allowed other technological advances.

Reviewer 3 Report

The authors addessed most of the my concerns. Thus, the paper is acceptable for publication.

Author Response

We would like to thank Reviewer for taking the time and effort necessary to review the manuscript.